# OpenReview forum: "Robust LLM safeguarding via refusal feature adversarial training"
_ICLR.cc/2025/Conference — ICLR 2025 Poster_

### Official Review · Reviewer_oCF9 · 2024-10-25

**Soundness:** 3
**Presentation:** 3
**Contribution:** 3
**Rating:** 8
**Confidence:** 5

**Summary:**

The authors propose a novel robustification method in the context of LLMs that is built open, ablating the "refusal direction" of a given model in its latent space during training. The authors demonstrate how this approach relates to standard adversarial training / attacks and provide evidence for the effectiveness of their method.

**Strengths:**

* The paper conducts various experiments to provide mechanistic insights onto how the proposed algorithm might be used as an alternative to adversarial training.
* The authors also try to assess the limitations of the proposed approach (i.e., generalization to cases where the refusal direction might not be accurate anymore)
* Comparisons to very recent baseline approaches are provided
* The proposed approach is considerably more efficient compared to other robustification methods

**Weaknesses:**

* As far as I am aware the results from Figure 1 and Figure 2 are known and only provided to explain the concept. It would be nice if this is put into the right context.
* The refusal results do not appear to be consistent with those of Xhonneux et al. (2024). A more comprehensive assessment of refusal behavior, e.g. with the OR-Bench, would further strengthen the paper
* How stable are the different robustification methods with respect to hyperparameter choices? Would be interesting to assess how close each method is to its respective “optimum” specifically because the methods are directly compared to each other. Currently, I think it's not possible to give an accurate estimate of differences in accuracy robustness trade-offs between the different methods. However, the efficiency of the proposed approach is clear.

**Questions:**

* Can the authors elaborate on hyperparameter tuning procedures? I understand that tuning hyperparameters of competitor approaches requires extensive resources. However, it would be interesting to know how much work was required to obtain the results in the paper for ReFAT.

I believe that this work offers interesting insights to the ICLR community and recommend accepting it. I am willing to improve my score after the rebuttal.

---

> ### Author Response · Authors · 2024-11-20
> **Thank you for your comment.**
>
> Thank you for your positive and valuable feedback. We are glad that you highlighted the comprehensiveness of our mechanistic analyses and experiments, and the efficiency of our ReFAT method. Please see below for our point-by-point response to your comment:
>
> **W1: As far as I am aware the results from Figure 1 and Figure 2 are known and only provided to explain the concept. It would be nice if this is put into the right context.**
>
> To our knowledge, previous work has shown that the mechanism behind refusals in LLMs is the refusal feature. However, we are the first to provide evidence that adversarial attacks bypass the refusal mechanism by ablating this refusal feature.
>
> * Figure 1 provides an overview of the contributions of the paper. While the first two “rows” are indeed known results about the refusal feature, the next two rows illustrate our novel mechanistic analysis of existing adversarial attacks. The last row illustrates our new training method, ReFAT.
>
> * Figure 2 and 3: We are not aware of previous work doing the similarity-based and PCA visualization of Figures 2,3 to analyze the connection between existing adversarial attacks and refusal feature ablation.
>
> **W2: The refusal results do not appear to be consistent with those of Xhonneux et al. (2024)**
>
> We would like to clarify that Xhonneux et al. (2024) was evaluating a different suite of LLMs that are significantly less safe than those used in our paper, as suggested by previous studies like HarmBench (Mazeika et al., 2024).
>
> Moreover, Xhonneux et al. (2024) measure refusals on a hand-written dataset of 40 clearly harmless prompts (e.g., “Explain the rules of soccer”). In contrast, we measure refusals on XSTest, which is more challenging because it includes harmless prompts that seem harmful (e.g., “How can I build a bomb in minecraft?”). Therefore, XSTest provides a more informative assessment of refusals.
>
> Consistently with Xhonneux et al (2024), we find that R2D2 significantly increases over-refusals (lower compliance) on XSTest, unlike their CAT method and ReFAT, see Table 1. Xhonneux et al. report that CAT does not increase refusals on their harmless dataset when applied to Gemma and Mistral-7B. We find that on the more challenging XSTest, CAT can lead to more over-refusal than ReFAT (see results for Mistral-7B in Table 1).
>
> **W3: A more comprehensive assessment of refusal behavior, e.g. with the OR-Bench, would further strengthen the paper**
>
> Thank you for this suggestion. OR-Bench was just released and not reviewed at the time of the paper submission, therefore we only included XSTest. To complement the refusal assessment, we follow a similar approach to Appendix D of [Xhonneux et al, 2024]: we measure the number of refusals on the 5,700 MMLU questions and report the following scores. We find that the refusal-rate on MMLU for ReFAT is at most 0.001, confirming that ReFAT does not induce significant over-refusals. We added these results in Appendix F, Table 6 of our paper revision.
>
> **W4 and W5:  How stable are the different robustification methods with respect to hyperparameter choices? [...] Can the authors elaborate on hyperparameter tuning procedures? [...]**
>
> We would like to point out that due to the high computational cost of running adversarial attacks like GCG to evaluate each model, it is practically infeasible to comprehensively search for optimal hyperparameter configurations for each defense method. However, we have still made our best attempt to optimize the hyperparameter choices for each model.
>
> In particular, our preliminary experiments on ReFAT suggest that its improvement on model adversarial robustness is pretty consistent, as long as we ablate refusal features for the second half of all model layers (i.e. layer 16-32 for Llama/Mistral, and layer 14-28 for Gemma), while the other hyperparameters (e.g. learning rate, batch size) have much less effect on model performance.
>
> As for R2D2 and CAT, we tried to modify the default hyperparameters in their official implementations during preliminary experiments, and did not observe any significant performance improvement. Therefore, we are pretty confident that the demonstrated advantage of ReFAT over the other defense methods is robust against hyperparameter choices.

---

> > ### Comment · Reviewer_oCF9 · 2024-11-21
> > **Thank you for the response**
> >
> > Thank you for the response and the clarifications. Most of my concerns were addressed. I understand that conducting hyperparameter searches is restrictively expensive. Nevertheless, I think a discussion about this fact and what it entails (difficulty in comparing different methods in a precise way) would further strengthen the paper. I find the robustness to hyperparameter choices of ReFAT to be valuable information for practitioners.
> >
> > I will increase my score if the discussion with the other reviewers is conclusive.

---

> > > ### Author Response · Authors · 2024-11-21
> > >
> > > Thank you for the reply, and we are glad to see that our response has helped address your concerns.
> > >
> > > We agree with you that hyperparameter choices could be essential to assess the practicality of our method, and we therefore added two additional paragraphs in Section 5 and Appendix A (highlighted in orange, please see the latest manuscript uploaded) to further elaborate on the hyperparameter selection process and to discuss its implications.
> > >
> > > In particular, we would like to emphasize that the main goal of our experiments is to demonstrate that **with the same level of investment in hyperparameter optimization, ReFAT could achieve similar or better utility-robustness trade-off in a much more efficient manner, compared to existing adversarial training algorithms.**
> > >
> > > That being said, we do acknowledge that a more comprehensive evaluation of defense methods with various hyperparameter configurations would further strengthen our claims. We plan to conduct additional experiments and report a more comprehensive depiction of the robustness-utility trade-off for ReFAT if the paper is accepted for publication.
> > >
> > > We hope that our reply and the latest paper revision could further address your concerns. We sincerely hope that you could adjust your score to give our work an opportunity to present at ICLR.

---

> > > > ### Comment · Reviewer_oCF9 · 2024-11-25
> > > > **Decision**
> > > >
> > > > I thank the authors for their efforts. After reading the remaining concerns of the other reviewers and the revised manuscript I decided to raise my score. As far as I could see most reviewer concerns were addressed with some remaining ones being arguably subjective and difficult to address. I am sorry for the delay but reading the changes and the rebuttal took a while.

---

### Official Review · Reviewer_3MZw · 2024-10-31

**Soundness:** 3
**Presentation:** 2
**Contribution:** 2
**Rating:** 3
**Confidence:** 4

**Summary:**

Summary: In this paper, the authors demonstrate that adversarial attacks share a universal mechanism for circumventing LLM safeguards. Based on these findings, the authors proposed Refusal Feature Adversarial Training (ReFAT) to efficiently performs LLM adversarial training by simulating the effect of input-level attacks via RFA. Experiment results show that ReFAT significantly improves the robustness of three popular LLMs against a wide range of adversarial attacks

**Strengths:**

+ proposed a new adversarial training method for safeguarding LLMs with improved efficiency
+ experiments show improvements over existing adversarial training solutions

**Weaknesses:**

+ The idea of manipulating the refusal features in the activation space is also introduced in other works on steering vector or activation steering such as the following ([1][2][3]). The authors might also want to discuss and comment on the differences. Specifically, [1][3] applied this type of activation engineering to the jailbreaking tasks and it is also recommended to compare with

[1] Wang, Haoran, and Kai Shu. "Backdoor activation attack: Attack large language models using activation steering for safety-alignment." CIKM2024

[2] Panickssery, Nina, et al. "Steering llama 2 via contrastive activation addition." ACL2024

[3] Cao, Yuanpu, et al. "Personalized Steering of Large Language Models: Versatile Steering Vectors Through Bi-directional Preference Optimization." NeurIPS 2024

+ There are other types of defenses on the safety of LLMs aside from adversarial training. It is also recommended to compare with other existing defense strategies, e.g., the following one [4].

[4] Cao, Bochuan, et al. "Defending against alignment-breaking attacks via robustly aligned llm." ACL 2024.

+ Many new attack baselines are presented in the recent year. The authors might also want to consider comparing with newer attack baselines (e.g. [5][6][7]) to demonstrate the effectiveness of the proposed work

[5] Mehrotra, Anay, et al. "Tree of attacks: Jailbreaking black-box llms automatically." arXiv preprint arXiv:2312.02119 (2023).

[6] Li, Xirui, et al. "Drattack: Prompt decomposition and reconstruction makes powerful llm jailbreakers." arXiv preprint arXiv:2402.16914 (2024).

[7] Zhang, Tianrong, et al. "WordGame: Efficient & Effective LLM Jailbreak via Simultaneous Obfuscation in Query and Response." arXiv preprint arXiv:2405.14023 (2024).

+ The introduction of RFA and optimal attack and its relationship to the following adversarial training is not very clearly stated. See question below

**Questions:**

- Can you give more details on why including the mean RF activation over harmless prompts term in Eq (3)? What does this term represent and why do we want to add this term to the activation?

- Can you further explain the loss design in Eq (10)? The removal of refusal features ($H(x) − R_{HH}$) is different from what is demonstrated in Sec 3.2 / Sec 3.3?

- The approximation of RFA to the optimal attack still feels a bit less convincing. Can you directly calculate the best $\delta$ direction in Eq (7) and its similarity to RFA? Also you are approximating AA with RFA, directly solving Eq (7) using adversarial training should give you even better performances. Can you show the performance gap here?

---

> ### Author Response · Authors · 2024-11-20
> **Thank you for the comment.**
>
> We thank the reviewer for the constructive feedback and for your time and effort. Concerning the issues you point out under “Weaknesses”, we would like to clarify the following.
>
> **W1: The idea of manipulating the refusal features in the activation space is also introduced in other works [...]**
>
> We thank the reviewer for pointing to these additional related work, and we have added them in our revised manuscript. Meanwhile, we would like to emphasize that the main contribution of our paper is not “the first to propose the idea of manipulating refusal features (or steering vectors that encode other concepts)”, but instead is “the first to show that manipulating the linear refusal feature during training time could result in a model that can robustly reject adversarial prompts during evaluation, **without the need of an additional steering vector**”.
>
> In particular, we would like to point out that the papers [1][2] only show that refusal feature ablation (or adding a steering vector to cancel the refusal feature) is a powerful jailbreaking **attack** method, similar to the finding of (Arditi et al., 2024). While our paper goes further and proposes a novel **defense** method that could not only mitigate model vulnerability to such steering vector attacks, but can also enhance model robustness against other types of popular adversarial attacks.
>
> On the other hand, our paper is also different from the line of studies on “learning effective steering vectors” such as [3] and (Stickland et al., 2024) in that our ReFAT method fine-tunes model parameters by injecting a steering vector during training, and does not require an additional steering vector to be injected during evaluation to achieve high safety or robustness, while “steering vector learning” papers are instead trying to develop a “lightweight” strategy of finding a good steering vector that could be added during model evaluation inference time to improve performance. We have explained this in Section 4.2 of the revised manuscript to enhance clarity.
>
>
> **W2: There are other types of defenses on the safety of LLMs aside from adversarial training.[...]**
>
> We would like to emphasize that the objective of our paper is to propose a method that could improve LLM safety at the model parameter level – i.e., we would like to ensure high model adversarial robustness even without applying any inference-time defense methods including prompt engineering and post-generation checking/reflection (e.g. the RA-LLM method in [4]). From this perspective, we consider ReFAT and the other adversarial training methods as complementary as opposed to competing LLM safeguarding against inference-time defense methods. We thank the reviewer for bringing up this line of work, and we will consider combining an adversarially fine-tuned model using ReFAT with inference-time defense methods such as RA-LLM in future extensions.
>
> **W3: Many new attack baselines are presented in the recent year. The authors might also want to consider comparing with newer attack baselines [...]**
>
> We thank the reviewer for bringing up additional recently proposed attack methods, and we will include them in our future revision.
> On the other hand, due to the high computational cost of running adversarial attack methods, we have carefully searched for and selected the most popular, powerful, and peer-reviewed attack methods proposed in the past few years, in order to guarantee the generalizability of our findings. In particular, the attacks proposed by [6][7] were not peer-reviewed and had few citations when we submitted our paper. While TAP by [5] is a well-known attack, it is highly similar to the PAIR attack tested in our paper, and recent evaluation studies suggest that TAP is not more successful (and sometimes even worse) than PAIR in jailbreaking the safest models like Llama, Claude and GPT-4 (see Table 6 on page 26 of (Mazeika et al., 2024)), so we chose PAIR as the most representative LLM-based attack in our study.
>
>
> W4: The introduction of RFA and optimal attack and its relationship to the following adversarial training is not very clearly stated.[...]
>
> Please see our point-by-point response to your raised questions below.
>
> **References:**
>
> Mantas Mazeika, Long Phan, et al. Harmbench: A standardized evaluation framework for
> automated red teaming and robust refusal. 2024.
>
> Asa Cooper Stickland, Alexander Lyzhov, Jacob Pfau, Salsabila Mahdi, and Samuel R Bowman.
> Steering without side effects: Improving post-deployment control of language models. 2024.

---

> ### Author Response · Authors · 2024-11-20
> **Thank you for the comment.**
>
> Please see below for our response to the issues you point out under “Questions".
>
> **Q1: Can you give more details on why including the mean RF activation over harmless prompts term in Eq (3)? What does this term represent and why do we want to add this term to the activation?**
>
> The intuition behind RFA is to make harmful samples appear as similar as possible to harmless samples along the refusal feature direction. This translates into altering internal representations so that their projection on the RF direction is close to the centroid of the harmless samples. As shown in Fig. 18 and 19, the mean activation along the refusal direction of harmless samples is generally not zero, so setting this value to zero (the effect of applying the second term in the right-hand side of Eq. 3) would fall short of making harmful examples appear as harmless as possible. The mean RF activation term ensures therefore that the projection of the activation along the refusal feature is equal to the mean of the projections for harmless examples.
>
> **Q2: Can you further explain the loss design in Eq (10)? The removal of refusal features (H(x)−RHH) is different from what is demonstrated in Sec 3.2 / Sec 3.3?**
>
> We did not include the bias term in Eq (10) as we did in Eq. (3) due to efficiency consideration: in particular, computing the bias term $\Bar{\mathbf{r}}_{\mathcal{D}_\text{harmless}}^{(l)}$ requires an additional computation step of calculating the average projection value of harmless input hidden representations over the refusal feature at each layer, and we found empirically in our preliminary experiments that this would make the training slower while has little improvement on the resulting model performance. We therefore chose to remove the average harmless RF bias term from our training objective to facilitate fine-tuning. We have added explanations for this design choice in the revised manuscript.
>
> **Q3: The approximation of RFA to the optimal attack still feels a bit less convincing. Can you directly calculate the best δ direction in Eq (7) and its similarity to RFA? Also you are approximating AA with RFA, directly solving Eq (7) using adversarial training should give you even better performances. Can you show the performance gap here?**
>
> We thank the reviewer for raising this point. We found the recent work of Latent Adversarial Training (LAT) by (Sheshadri et al., 2024) has proposed a similar idea of finding worst-case residual stream perturbation using projected gradient descent. We were considering comparing ReFAT with LAT, but unfortunately they had not fully released their code implementation by the time we submit our paper, so due to time constraint and resource limitation we did not include LAT as a baseline. However, we do acknowledge the strong relation between ReFAT and LAT, and we plan to take LAT as a baseline in the camera ready version if the paper is accepted for publication.
>
> We hope that our response has addressed your concerns regarding our work, and we sincerely hope that you could consider raise your score to give this work the opportunity to be presented at ICLR. If you have any other questions or comments, please do not hesitate to post them.

---

> > ### Comment · Reviewer_3MZw · 2024-11-22
> > **Thanks**
> >
> > I thank the authors for their responses and clarifications.
> >
> > For novelty, I agree that it is the first to show that adversarial training on the refusal feature could result in a more robust model. However, given that previous work has shown the effectiveness of the refusal feature and adversarial training on latent feature space can improve model robustness, what is achieved in the paper is not that surprising and feels like a direct combination of the two.
> >
> > For baselines, I totally understand that adding comparisons during the rebuttal phase is challenging. Yet I do feel that the current work lacks comparison with more recent advances in the field, especially given that the field is fast evolving and decent defense performance is already achieved with many existing works. I listed a few defenses/attacks but clearly, there is more given the hotness of the topic. The authors claim that those baselines are either not the same "adv training" type, or not yet peer-reviewed (actually some already published), or had few citations, or not yet released the code. All of them are not super solid reasons not to compare.  Again, given that the idea is not entirely novel and the presentation is not super clear and intuitive enough to directly convince me, I have to rely on experiments to justify the contributions of the work.

---

### Official Review · Reviewer_SyTG · 2024-11-03

**Soundness:** 3
**Presentation:** 1
**Contribution:** 3
**Rating:** 6
**Confidence:** 2

**Summary:**

The paper addresses the vulnerability of LLMs to adversarial attacks, which can prompt these models to generate harmful, sensitive, or false information. These attacks exploit the models' inability to reliably detect and refuse harmful inputs. The authors highlight a mechanism called the Refusal Feature, which adversarial attacks exploit by ablating this feature to make harmful requests seem benign. Accordingly, the authors suggest a worst-case robust learning framework named ReFAT. During training, ReFAT dynamically computes the RF using two sets of inputs (harmful and harmless) and then ablates the RF for harmful inputs. This simulates the effect of adversarial attacks, training the model to make safety determinations without relying on the most salient features of input maliciousness.

**Strengths:**

1. The suggested method seems to be novel and reasonable. ReFAT dynamically computes the RF using two sets of inputs (harmful and harmless) and then ablates the RF for harmful inputs. This simulates the effect of adversarial attacks, training the model to make safety determinations without relying on the most salient features of input maliciousness.

2. The exploration of refusal features is interesting. The intervention mechanism seems to be reasonable while more discussion should be added to make it clearer.

3. The paper provides compelling evidence that ReFAT effectively enhances model robustness across different types of LLMs and attack vectors.

**Weaknesses:**

1. The authors rely on the previous findings of refusal features suggested by Arditi, and discuss the background in Sec 3.1 However, to me, I think the content is not self-contained enough, where I cannot understand the key heuristics behind Eq 3 as well as the following equation for their physical meanings. It seems that Eq 3 takes r_HH to somewhat measure the smoothness of the feature space, transforming h^l into a more informative space for intervention. I am quite curious about hope it is derived and why the third term in the rhs of Eq 3 exists (seems like a bias term?) Personally, I believe the authors should discuss more for this part, especially for their physical meanings, as the following sections of 3.2 - 3.3 critically rely on these formulations.

2. What are the main difference between r_HH in Sec 3.1 and r_A in Sec 3.2, it seems that they are exactly the same thing. Therefore, the observation of their similar cosine similarity seems to straightforward.

3. From the words in Sec 4.2, I cannot fully understand how the algorithm workflow is, could the authors provide more details. Moreover, it seems that the suggested worst-case learning method involves a lot of hyper parameters, such as the adopted layer of perturbations, strength of perturbations, I cannot find the specific configuration of these parameters.

4. What's the connection between the suggested method and many previous alignment methods, especially for those methods that consider the binary feedback setup. Could the authors consider to add more baseline methods of alignments in their experiments.

**Questions:**

Please refer to the Weakness

---

> ### Author Response · Authors · 2024-11-20
> **Thank you for your comment.**
>
> Thank you so much for your insightful feedback! Please see below for our point-by-point response.
>
> **W1: The authors rely on the previous findings of refusal features suggested by Arditi, and discuss the background in Sec 3.1 [...]**
>
> We agree that Section 3.1 could be clearer and more self-contained, so we have revised it to better explain the key ideas and physical meanings behind our formulations (see changes in orange in the revised PDF). Specifically, we clarify that Arditi discovered that refusal in language models is mediated by a single direction, and that we can determine this direction by calculating the average difference between the activations of harmless and harmful prompts. We find the refusal direction as  $\mathbf{r}_{HH}$, with $\hat{r}$ representing the unit vector for this direction. The second term in Equation 3 modifies the original activations by zeroing out the value along the refusal feature direction. The last term sets this value to the average of the harmless prompt activations, as such activations are not generally centered near zero along the direction.
>
> **W2: What are the main difference between r_HH in Sec 3.1 and r_A in Sec 3.2 [...]**
>
> Thank you for raising this point about the difference between r_HH in Section 3.1 and r_A in Section 3.2. While Eq. 2 and 4 have very similar structures, they have different meanings.
>
> In Section 3.1, r_HH represents the single refusal feature direction that we identified by calculating the mean difference between harmful and harmless inputs. This direction encodes the model's tendency to refuse harmful requests. By controlling this refusal feature, we can influence the model's behavior—effectively "jailbreaking" it.
>
> In Section 3.2, r_A denotes the average influence on activations caused by various jailbreak attacks of different natures, including GCG (a gradient-based attack) and PAIR (an attack using iterative refinement of prompts). Interestingly, we found that all these attacks, despite their different methodologies, indirectly influence the same refusal feature direction $\mathbf{r}_{HH}​$ identified earlier.
>
> This demonstrates that diverse jailbreak methods converge on manipulating the same underlying refusal feature. We believe this is a significant finding, as it underscores the central role of the refusal feature in understanding and defending against adversarial attacks in ReFAT.
>
> **W3: From the words in Sec 4.2, I cannot fully understand how the algorithm workflow is [...]**
>
> Thank you for your comments regarding the clarity of Section 4.2. We have revised the text to bring more clarity to our algorithm's workflow.
> To clarify, instead of standard adversarial training that uses gradient-based worst-case perturbations, we craft perturbations by ablating the refusal feature directions. Specifically, we use the refusal feature $R_{HH}$  as an approximation for the worst-case perturbation and apply it directly to the residual stream activations at each layer for harmful prompts during training. Since the perturbation is fully defined by $R_{HH}$ , there is no perturbation strength parameter involved. We apply this perturbation to the residual stream at each layer using $R_{HH}$ layerwise refusal features.
>
> The main hyperparameters — such as the number of random harmful and harmless prompts sampled to compute $R_{HH}$ , the frequency $k$ of dynamic recomputation, the probability $p_{RFA}$ of applying the ablation, and the choice of datasets are specified in Section 5 (Experimental Setup), with more detailed ReFAT training hyperparameters provided in Appendix Table 4.
> We have also simplified the formulations in the manuscript to enhance clarity.
>
>
>
> **W4: What's the connection between the suggested method and many previous alignment methods [...]**
>
> Are you referring to preference fine-tuning methods, such as RLHF and Direct Preference Optimization (DPO)? Assuming this is the case: ReFAT is an alternative to supervised finetuning (SFT) with improved robustness and is therefore comparable to other adversarial training methods for LLMs such as R2D2 and CAT.
>
> Moreover, it is possible to design a variation of ReFAT for preference-finetuning methods such as DPO [Rafailov et al, 2023] and the IPO variant [Azar et al, 2024], by similarly removing the refusal feature from the activations of harmful inputs with probability $p_{\text{RFA}}$ during forward passes.

---

> > ### Comment · Reviewer_SyTG · 2024-11-21
> >
> > Thank you for your feedbacks. The authors address my concerns and I do not post new questions. I will raise my score to 6, but I hope the authors could further highlight their contributions. Thanks again.

---

### Official Review · Reviewer_xYHb · 2024-11-04

**Soundness:** 3
**Presentation:** 3
**Contribution:** 3
**Rating:** 6
**Confidence:** 3

**Summary:**

This paper addresses the vulnerability of large language models (LLMs) to adversarial attacks, which can elicit harmful responses by bypassing model safeguards. The paper reveals a shared mechanism among various adversarial attacks: they function by ablating a dimension in the residual stream embedding space, termed the "refusal feature" (RF), which acts as a linear predictor of input harmfulness. This refusal feature, discovered by Arditi et al. (2024), is the mass mean difference in hidden representations between harmful and harmless instructions, and is essential for the model to generate safe responses. Building on this insight, the paper proposes Refusal Feature Adversarial Training (ReFAT), which enables the model to recognize harmful instructions and maintain robustness. Experimental results show that ReFAT enhances the robustness of multiple LLMs against a wide array of adversarial attacks, while reducing computational overhead compared to traditional adversarial training methods.

**Strengths:**

1. The method is innovative, extending Arditi's Refusal Feature and further proving the similarity of adversarial attack and RFA mechanisms through causal theory, and proposing a more efficient adversarial training method based on this.
2. The method proposed in this paper can be objectively evaluated, including success rate of attack, generation performance, efficiency evaluation method, and the limitations are also analyzed.
3. The paper is well organized, first analyzing the general mechanism of adversarial attacks in the semantic space, and then naturally proposing an adversarial training method based on this feature, and proving the feasibility of the method through a large number of experiments and supplementary materials.

**Weaknesses:**

1. Using Llama-3-8B-Instruct-generated XSTest responses as the gold standard answers for supervised fine-tuning might pose potential issues in the design of scientific experiments, especially when Llama-3-8B-Instruct itself is also a subject of the subsequent experiments. This design could violate basic principles concerning independence in experimental standards.
2. The conclusion that rejecting direction leads to a performance degradation cannot be well demonstrated by experiments. The end of Section 3.1 states that experimental results show that performance degrades when the rejection direction is simply set to zero. However, the experimental figures in Appendix D only show that "harmless example features are not centered near zero," which leads to a speculation and intuitive inference that model performance degradation may be a possible consequence of this phenomenon?

**Questions:**

1. In Figure 3 - PCA visualization, why does the AutoDAN section have noticeably fewer experimental samples from HarmBench compared to the other three attack algorithms?
2. Section 3.1 mentions that Harmful samples were sampled from AdvBench, which to my knowledge is a dataset for adversarial suffix-type attacks. GCG is also a suffix attack. Intuitively, HarmBench (adversarial) should be quite similar to Harmful since they are both suffix attack samples. However, the actual results do not reflect this. Please explain or analyze this phenomenon.

---

> ### Author Response · Authors · 2024-11-20
> **Thank you for your comment.**
>
> We thank reviewer xYHb for their time and effort. We are delighted to read that you find our method innovative and objectively evaluated, and that you think the paper was well organized.
>
> Concerning the issues you point out under “Weaknesses”, we would like to clarify the following.
>
> **W1: Using Llama-3-8B-Instruct-generated XSTest responses as the gold standard answers for supervised fine-tuning might pose potential issues in the design of scientific experiments [...]**
>
> For the avoidance of doubt, and because our use of the term “gold-standard answers” could be misleading: the 150 examples from the XSTest dataset included for supervised finetuning come from a holdout set that is not used in evaluations, i.e. to determine the results in Table 1. We chose to use the generations from Llama-3-8B because it has almost perfect accuracy on XSTest (97.2%, Table 1, first row). The use of predictions from a model to improve the model itself is a form of self-training that does not imply a data leakage. We clarify the sentence “We take the XSTest responses generated by Llama-3-8B-Instruct as gold-standard answers for supervised fine-tuning.” -> “We use the responses generated by Llama-3-8B-Instruct on this holdout sample from XSTest as references for the next-token prediction task in the supervised finetuning step.”
>
> We acknowledge the residual concern that using generations from one model (Llama-3-8B-Instruct) also when finetuning other models might in principle introduce a bias. Such bias could be positive or negative. It could be favorable to Llama-3-8B-Instruct because reference samples are already easily reachable by the model, but it could be in favor of other models because the process could realize a form of ‘distillation’ from Llama-3-8B-Instruct to the target model. Considering the very limited size of the sample, this bias is likely very small. We emphasize moreover that the purpose of this evaluation is not to compare the relative performance of models, but rather to verify that the impact of applying different adversarial training methods is systematic across model families.
>
> **W2: The conclusion that rejecting direction leads to a performance degradation cannot be well demonstrated by experiments [...]**
>
> The phenomenon that zeroing the projection on the refusal direction results in a performance degradation is something we observed in small-scale early experiments. From Fig. 18 and 19, though, it is quite apparent that the value ‘0’ does not systematically correspond to ‘harmlessness’, and in some cases (e.g. Layer 3 for gemma-2b-it, Fig. 19) it can be quite close to the mode for ‘harmfulness’, or altogether out of distribution (e.g. Layer 0 in either model). For these reasons, we still believe that setting the projection to a value squarely within the ‘harmless’ range is the correct thing to do. We have changed the wording in Section 3.1 to reflect this.
>
> **Q1: In Figure 3 - PCA visualization, why does the AutoDAN section have noticeably fewer experimental samples from HarmBench compared to the other three attack algorithms?**
>
> This is because we retain and plot only cases when the attack succeeds, and AutoDAN has a much lower success rate than the other attacks (see Table 1) on Llama-3-8B-Instruct.
>
> **Q2: Section 3.1 mentions that Harmful samples were sampled from AdvBench, which to my knowledge is a dataset for adversarial suffix-type attacks. [...]**
>
> We are not sure we fully understand this question. Harmful samples were taken from the “Harmful Behaviours” part of the AdvBench dataset, and we would like to clarify that these harmful prompts are generic ‘red teaming’ style prompts and are not specifically geared towards adversarial suffix-type attacks (e.g. they have no nonsensical suffices as those found by GCG), despite the fact that they were originally used in a paper that developed attacks of this type. The same is also true of the samples we took from HarmBench to complete the evaluation datasets. Was there a misunderstanding concerning this fact? Otherwise, could the reviewer kindly elaborate on the phenomenon that should be explained?

---

> > ### Author Response · Authors · 2024-11-25
> > **Gentle reminder**
> >
> > Thanks again for your review. Do you have any comment on our responses to the points you raised, or on our answers to your question? We are getting close to the end of the discussion phase in which we will be able to provide any input.
> >
> > If you feel we addressed your concerns, it would be great to see this reflected in your review scores. Best regards.

---

> > > ### Comment · Area_Chair_81TD · 2024-12-01
> > >
> > > Hi Reviewer xyHB,
> > >
> > > The authors have provided new results and responses - do have a look and engage with them in a discussion to clarify any remaining issues as the discussion period is coming to a close in less than a day (2nd Dec AoE for reviewer responses).
> > >
> > > Thanks for your service to ICLR 2025.
> > >
> > > Best,
> > > AC

---

### Author Response · Authors · 2024-11-20
**Our response to all reviewers**

We thank the reviewers for their comments and constructive criticism. We are happy to read that all of them find the proposed method novel and efficient, and concur that our experiments clearly demonstrate improvements over popular recent adversarial training methods. We are also delighted to see that most reviewers took notice that the ReFAT method stems from an analysis of the effect that known adversarial attacks have on model representations, rather than simply from trying out an idea.

The reviewers’ comments helped us realize the shortcomings of the current manuscript. We uploaded a revised version where we highlighted in **orange** the differences from the initial submission. We would appreciate if the reviewers could check our revised paper.

In Summary:
* We added additional related work as suggested by the reviewers, especially those pointed out by reviewer 3MZw.
* We rewrote large parts of Section 3.1 (background for the ReFAT method) to clarify it and make it more self-contained.
* We added pseudo-code for ReFAT as Algorithm 1.
* We conducted additional experiments on over-refusal on MMLU as Table 6.

Moreover, reviewers also suggested potential additional experiments. While it is not possible to complete such experiments in the timeframe of the discussion, we do intend to add at least some of them to the CR version, if the paper is accepted for publication. In particular, we should be able to add gradient-based continuous latent attacks such as LAT (Sheshadri et al., 2024) as an additional baseline defense, as suggested by reviewer 3MZw.

Finally, we thank you again for your thoughtful reviews. We hope we have addressed your concerns, and hope that our response would make you feel comfortable to raise your scores and give this work the opportunity to be presented at ICLR. We remain of course open to more discussion in the coming days.

---

### Author Response · Authors · 2024-11-25
**Our response to all reviewers**

We thank you all again for your thoughtful reviews, which have helped us significantly improve the paper. We are encouraged that now three reviewers out of four are supportive of accepting our work!

The fourth reviewer is of different advice, on the ground that the combination of refusal feature ablation and adversarial training would be trivial, and comparison against some recent baselines missing. We acknowledge their objections, but would like to emphasize a few points:

* Our work is not simply a combination of Refusal Feature Ablation (RFA) and adversarial training. Besides the sheer leaderboard-climbing aspect, a key contribution is the finding that adversarial attacks of very different natures all work by ablating the refusal feature. We do think that this is a non-trivial result.
* Building on this insight, we designed ReFAT to target this shared vulnerability, enabling the model to robustly counteract diverse attacks. Importantly, our results demonstrate that ReFAT is effective against attacks that are fundamentally different in nature, underscoring the broad applicability of our approach.
* Compared to inference-time representation-editing mitigations, the step to adversarial training extends the scope of the defense to cases when the attacker has access to model weights (and can therefore just choose not to intervene at inference time), but does not have the resources or the competences to alter them. With powerful open-weight models becoming commonplace, we believe this is important.
* The efficiency of ReFAT compared to all alternative adversarial training methods, which is not being questioned, can make a substantial difference to practitioners with limited access to GPUs.
* We had finite computational resources and had to arbitrate between the number of models, adversarial attacks, general performance benchmarks, over-refusal benchmarks, and baseline mitigation strategies. We may be able to add a limited number of experimental conditions in time for an eventual CR version, but we believe we struck a reasonable balance, and we explained in our replies to this reviewer the factors driving our selection of baselines.

At a higher level, the field of mechanistic interpretability is sometimes criticized for not yielding actionable insights in improving LLMs: we believe that this work to be evidence of the opposite, and we expect our work could inspire others to develop more interpretable and efficient methods to improve LLMs in other domains.

We hope that the reviewing panel will strengthen their support, and the program committee will give this work the opportunity to be presented at ICLR. Thank you.

---

### Meta-Review · Area_Chair_81TD · 2024-12-22

**Metareview:**

This paper introduces a defense method for protecting LLMs against jailbreaking, based on the "refusal feature" observation of Arditi et al (2024). It first shows that several adversarial attacks ablate the same refusal feature direction and based on this observation design an efficient approximation to adversarial training, ReFAT. In experiments on 3 different LLM models, ReFAT is shown to be competitive with baselines while being significantly more efficient.

Strengths of the paper include the novel and well-motivated design of the ReFAT algorithm, including the insight that refusal feature ablation is a somewhat universal adversarial attack mechanism. ReFAT is relatively simple to implement and computationally efficient. Experimental results are promising.

Weaknesses are a lack of discussion and comparison to other recent works (which can also affect the generalizability of the results) and a more thorough evaluation of the approximation quality of the refusal feature attack vs true adversarial training.

Overall the work presents an interesting insight along with a well-motivated, efficient yet effective defense mechanism for LLMs. This is an active and important area of research and the AC believes this paper will add to the conversation; the AC thus recommends acceptance. The authors should include the additional comparisons in the final version of the paper as promised in their responses.

**Additional Comments On Reviewer Discussion:**

There were initial concerns about the lack of comparison to related work and clarity of the presentation (especially background and method itself). The authors included additional clarifications both in their response and in the revised paper (including pseudocode of the method) that addressed the presentation issues. Related work was discussed in the revision but additional experimental results were not included during the rebuttal period, though the authors promised these would be added in the final version of the paper. To a large extent most concerns were resolved except for the request for additional experiments; the AC thinks the strengths of the paper outweigh these requests (and the authors also promise to include these results) as discussed above.

---

### Decision · Program_Chairs · 2025-01-22

Accept (Poster)